# Identification of a Novel *FOXP1* Variant in a Patient with Hypotonia, Intellectual Disability, and Severe Speech Impairment

**DOI:** 10.3390/genes14101958

**Published:** 2023-10-18

**Authors:** Mario Benvenuto, Pietro Palumbo, Ester Di Muro, Concetta Simona Perrotta, Tommaso Mazza, Giuseppa Maria Luana Mandarà, Orazio Palumbo, Massimo Carella

**Affiliations:** 1Division of Medical Genetics, Fondazione IRCCS Casa Sollievo della Sofferenza, San Giovanni Rotondo, 71013 Foggia, Italy; m.benvenuto@operapadrepio.it (M.B.); p.palumbo@operapadrepio.it (P.P.); e.dimuro@operapadrepio.it (E.D.M.); o.palumbo@operapadrepio.it (O.P.); 2Dipartimento Degli Studi Umanistici, Università Degli Studi di Foggia, 71122 Foggia, Italy; 3Medical Genetics Unit, Maria Paternò Arezzo Hospital, 97100 Ragusa, Italy; concettasimona.perrotta@asp.rg.it (C.S.P.); luana.mandara@asp.rg.it (G.M.L.M.); 4Unit of Bioinformatics, Fondazione IRCCS Casa Sollievo della Sofferenza, San Giovanni Rotondo, 71013 Foggia, Italy; t.mazza@operapadrepio.it

**Keywords:** *FOXP1*, targeted resequencing, neurodevelopmental disorders

## Abstract

The FOXP subfamily includes four different transcription factors: FOXP1, FOXP2, FOXP3, and FOXP4, all with important roles in regulating gene expression from early development through adulthood. Haploinsufficiency of *FOXP1*, due to deleterious variants (point mutations, copy number variants) disrupting the gene, leads to an emerging disorder known as “*FOXP1* syndrome”, mainly characterized by intellectual disability, language impairment, dysmorphic features, and multiple congenital abnormalities with or without autistic features in some affected individuals (MIM 613670). Here we describe a 10-year-old female patient, born to unrelated parents, showing hypotonia, intellectual disability, and severe language delay. Targeted resequencing analysis allowed us to identify a heterozygous de novo *FOXP1* variant c.1030C>T, p.(Gln344Ter) classified as likely pathogenetic according to the American College of Medical Genetics and Genomics guidelines. To the best of our knowledge, our patient is the first to date to report carrying this stop mutation, which is, for this reason, useful for broadening the molecular spectrum of *FOXP1* clinically relevant variants. In addition, our results highlight the utility of next-generation sequencing in establishing an etiological basis for heterogeneous conditions such as neurodevelopmental disorders and providing additional insight into the phenotypic features of *FOXP1*-related syndrome.

## 1. Introduction

The FOXP protein family (FOXP1, FOXP2, FOXP3, and FOXP4) is a group of transcription factors having important functions in several biological processes, such as embryological, immunological, and expressive language development [1].

The *FOXP1* gene (forkhead box P1) (OMIM #605515), located on 3p14.1, encodes for a transcriptional repressor protein involved in regulating the development of the brain and other organs. In particular, in the nervous system, *FOXP1* is involved in several important pathways required for proper brain development and function [2].

The closest homolog to *FOXP1*, and the best-known member of the FOXP family, is *FOXP2,* which was the first gene to be linked with an inherited monogenic form of language disorder. It was first reported when studying a three-generation pedigree in which severe speech delay was transmitted in an autosomal dominant manner [3]. Subsequently, several sporadic and familial cases of speech disorders associated with *FOXP2* mutations have been reported, corroborating the role of this gene in the etiology of developmental language delay [4]. From a functional point of view, FOXP1 and FOXP2 form heterodimers for gene expression regulation, and it has been suggested that they cooperate in common neurodevelopmental pathways through the co-regulation of shared targets [5].

*FOXP1* is also expressed in the developing striatal projection neurons and basal ganglia [5,6,7], and a reduction in the striatum has been observed in *FOXP1* haploinsufficient mouse models [8], while enlargement of the lateral ventricles has been observed in human patients with *FOXP1* haploinsufficiency [9]. Finally, *FOXP1* is necessary for proper development of motor circuits, and a lack of *FOXP1* in motor neuron progenitor populations may result in abnormal development of motor skills [10].

Haploinsufficiency of *FOXP1*, due to genetic variants that disrupt the gene, has recently been associated with a new syndromic form of neurodevelopmental disorder characterized, from a clinical point of view, by global developmental delay (DD), intellectual disabilities (ID), speech deficits, mild craniofacial dysmorphisms, and autism spectrum disorder (ASD) [11].

Notably, all of the causative *FOXP1* variants reported to date in which the inheritance pattern could be evaluated occurred de novo, corroborating the hypothesis that de novo mutations represent the major genetic cause of severe forms of sporadic neurodevelopmental disorders, such as ID, epilepsy, and autism [11,12].

Here, we describe a female patient with hypotonia, cognitive impairment, and severe speech delay in which we have employed targeted resequencing to detect a novel nonsense variant in *FOXP1*.

## 2. Materials and Methods

### 2.1. Genomic DNA Extraction and Quantification

Genomic DNA of the proband and her parents was isolated from peripheral blood leukocytes (PBL) by using Bio Robot EZ1 (Quiagen, Solna, Sweden). The quality of nucleic acid was tested by electrophoresis on a 1% agarose gel, and a Nanodrop 2000 C spectrophotometer (Thermo Fisher Scientific, Waltham, MA, USA) was used to determine its concentration. The family provided written informed consent to molecular testing and to the full content of this publication, and the study was approved by the Casa Sollievo della Sofferenza Hospital ethics committee (protocol no. 177CE).

### 2.2. Next Generation Sequencing Analysis

Targeted resequencing (TRS) on proband DNA was performed by a SureSelect gene panel (Agilent Technologies, Boulder, CO, USA) designed to enrich conserved coding regions of 338 genes known to be associated with syndromic and not syndromic forms of neurodevelopmental disorders.

Libraries were prepared following the manufacturer’s instructions using the SureSelect target enrichment kit (Agilent Technologies, Boulder, CO, USA). Next, targeted fragments obtained were sequenced on a NextSeq 500 sequencer (Illumina, San Diego, CA, USA) using the NextSeq 500 mid-output kit V2.5 (300-cycle flow cell).

Reads were aligned to the GRCh37/hg19 reference genome by Burrows-Wheeler Aligner (BWA) (v.0.7.17), while BAM files were sorted by SAMtools (v.1.7) and purged using Mark Duplicates from the Picard suite (v.2.9.0). Mapped reads were locally realigned using GATK 3.8. Reads with mapping quality scores lower than 20 or with more than one-half nucleotides with quality scores less than 30 were filtered out. Variants were identified using the GATK’s Haplotype Caller tool [13] and annotated based on frequency, impact on protein, conservation, and expression using distinct tools, as appropriate (dbSNP, ANNOVAR, EVS, 1000 Genomes, GnomAD, ESP, KAVIAR, and ClinVar) [14,15,16,17,18], and precomputed pathogenicity predictions were retrieved with dbNSFP v 3.0 (PolyPhen-2, SIFT, MutationAssessor, FATHMM, LRT, and CADD) [19] and evolutionary conservation measures.

Next, variant prioritization was performed. First, benign variants were excluded. Furthermore, the remaining variants were classified based on their clinical impact as pathogenic, likely pathogenic, or variants of uncertain significance according to the following criteria: (i) nonsense/frameshift variants in genes previously described as disease-causing by haploinsufficiency or loss-of-function; (ii) missense variant located in a critical or functional domain; (iii) variant affecting canonical splicing sites (i.e., −1 or −2 positions); (iv) variant absent in allele frequency population databases; (v) variant reported in allele frequency population databases, but with a minor allele frequency (MAF) significantly lower than expected for the disease (<0.002 for autosomal recessive disease and <0.00001 for autosomal dominant disease); (vi) variant predicted and/or annotated as pathogenic/deleterious in ClinVar and/or LOVD.

Confirmation and segregation analysis of the putative pathogenic variant were performed by Sanger sequencing on the proband’s and parents’ DNA. PCR products were sequenced using the BigDye Terminator v1.1 Sequencing Kit (Applied Biosystems, Foster City, CA, USA) and the ABI Prism 3100 Genetic Analyzer (Thermo Fisher Scientific). The clinical significance of the identified variants was interpreted according to the American College of Medical Genetics and Genomics (ACMG) guidelines [20]. Variant analysis was carried out considering the ethnicity of the patient.

Nucleotide variant nomenclature follows the format indicated in the Human Genome Variation Society (HGVS, http://www.hgvs.org) recommendations (accessed on 2 March 2023).

## 3. Results

### 3.1. Clinical Description

The patient was a 10-year-old female referred to the genetic clinic for hypotonia, intellectual disability, and severe language delay. She is the first child of healthy, non-consanguineous parents of self-reported European ancestry. Family anamnesis was unremarkable, with no history of congenital anomalies or ID/NDD. The delivery was vaginal at 37 weeks due to preterm premature rupture of the membranes. Birth weight of 3000 g (64th percentile), length of 49 cm (72nd percentile), and head circumference of 35.5 cm (96th percentile). Parents reported early developmental delays, and the patient was brought for a first evaluation at 11 months. At physical examination, she showed craniofacial dysmorphisms, including a prominent forehead, deep-set eyes, strabismus, an asymmetrical nasal tip, and raised ear lobes. In addition, she showed bilateral clinodactyly of the fourth finger of the hands, while echocardiography detected an ostium secundum atrial septal defect that was surgically corrected. At the age of 16 months, the individual demonstrated the ability to sit independently. By 2.8 years old, she exhibited limited verbal communication with words lacking semantic meaning and also started her first steps without external support. A new clinical evaluation was performed at 4 years. At that time, weight was 14 kg (10th percentile), length was 97.5 cm (25th percentile), and head circumference was 50.4 cm (50–75th percentile). Cognitive impairment (Stanford-Binet test) [21], language (PVCL test), and motor delay were predominant. In particular, TVL (Language Valuation Test, preschool level) is slightly below 3 years, the receptive vocabulary at TLR (Receptive Language Test) reaches the 5th percentile, and lexical production at TFL (Phono-Lexical Test: Assessment of lexical skills in preschool age) is at the 5th percentile [22]. In addition, exaggerated cupid’s bow, flat face, anteverted pavilions, widely spaced teeth, and genu valgum were added to the first clinical evaluation. At the last clinical evaluation, performed at 10 years old, weight 30.6 kg (25th percentile), length 135.5 cm (25–50th percentile), and head circumference 53.3 cm (75–90th percentile) were documented. Psychomotor development was severely delayed, as she showed language delay (dysarthria), attentional and motor instability associated with stereotypes, and involuntary movements of the upper limbs.

### 3.2. Genetic Analysis

TRS of the proband identified a novel variant c.1030C>T, p.(Gln344Ter) in the exon 9 of the *FOXP1* gene (NM_032682.6). The variant was detected with a depth coverage greater than 150x and good quality scores (Phread quality > 3000 and genotype quality = 99). This nonsense variant causes a premature stop codon and was absent in EP6500, dbSNP, GnomAD, and our in-house controls. Bioinformatics details are reported in Table 1. The mutation detected was classified as likely pathogenic following the ACMG guidelines and reported in the Leiden Open Variation Databases (LOVD) (https://databases.lovd.nl/shared/variants/0000920404#00001788) (accessed on 2 March 2023). We did not perform any functional evaluation of the variant at the protein level since it is a nonsense variant likely to result in nonsense-mediated decay, with no truncated protein likely to be expressed and detected in the patient’s cells. Segregation analysis was performed in the family. The primers were designed using Primer3.0 (http://bioinfo.ut.ee/primer3-0.4.0) (accessed on 1 January 2021) (*FOXP1*, exon 9, Forward primer: GCATTAAAGGGTGGGGATGG; *FOXP1*, exon 9, Reverse primer: AGGACAATGACAGGTTTTGGAC) and the polymerase chain reaction (PCR) was performed under standard conditions. The PCR products (371 bp) were sequenced on an ABI 3500xL DNA Analyzer (Applied Biosystems, Foster City, CA, USA). Parental DNA analysis showed that it was a de novo event (Figure 1).

## 4. Discussion

The *FOXP1* gene, located on chromosome region 3p14.1, encodes for a transcriptional factor that regulates gene expression in eukaryotes through transcriptional repression mechanisms. The FOXP subfamily is comprised of four genes: *FOXP1*, *FOXP2*, *FOXP3*, and *FOXP4* [11].

In this study, we report a female patient with a clinical diagnosis of neurodevelopmental disorder, craniofacial dysmorphisms, and bilateral clinodactyly of the 4th finger. To discover the underlying genetic cause of this syndromic phenotype, we performed TRS, which allowed us to reveal a novel heterozygous *FOXP1* variant, c.1030C>T, p.(Gln344Ter). This variant was not found in the parents, which was consistent with de novo inheritance.

Loss of function (LoF) variants in *FOXP1* lead to an emerging condition termed *FOXP1* syndrome (FOXP1S).

Our patient had a clinical phenotype that was typical of the condition and included many characteristics that had already been mentioned in earlier reports.

In a study conducted by Wang et al. (2004), the researchers provided evidence for the involvement of Foxp1 in the development of the murine heart [23]. Subsequently, Chang et al. (2013) discovered that FOXP1 exhibited higher expression levels in the fetal human heart compared to other fetal human tissues [24]. In our case, an ostium secundum atrial septal defect was found and fixed through surgery.

Lozano et al. reported a description of 62 FOXP1S-affected patients (41 male and 21 female) between 4 months and 31 years of age, 59 of whom included dysmorphology evaluations. The most frequently observed clinical features were a prominent forehead (48/59; 81%), a short nose with a broad tip or base (41/59; 69%), down-slanting palpebral fissures (24/59; 41%), ptosis (22/59; 37%), thick vermillion (18/59; 31%), ocular hypertelorism (17/59; 29%), and frontal hair upsweep (16/59; 27%). Less common clinical features included: single palmar crease (14/56; 25%), clinodactyly (13/56; 23%), pointed chin (12/57; 21%), high-arched palate (10/59; 17%), malformed ears (10/59; 17%), macrocephaly (9/59; 15%), and broad nasal bridge (8/59; 14%) [2].

During the same year (2021), Braden et al. reported an interesting study on twenty-nine patients affected by FOXP1S (males 12 and females 17), ranging in age from 2 years and 7 months old to 33 years and 4 months old, in order to delineate the speech and language phenotypes observed. These data allowed the authors to define a complex speech and language phenotype in which dysarthric and apraxic features seem to be the core traits [25].

Our patient shares some features with those previously described, including some aspects of the neurobehavioral phenotype, i.e., intellectual disabilities, hypotonia, and severe speech impairment (dysarthria), as well as facial dysmorphisms, i.e., a flat midface, widely spaced teeth, and malformed ears. For this clinical evidence, the description of our patient contributes to corroborating the core phenotype of this emerging syndrome.

In addition, our patient showed some manifestations rarely annotated in individuals with *FOXP1*-related phenotypes, including genu valgum and absence of sphincter control for up to 5 years. Cesaroni et al. (2023) [26] are the first researchers to describe a partial loss of sphincter control in a patient who was affected by FOXP1S. Our case is the second one to show this feature, and for this reason, it is helpful in evaluating how common this characteristic is. Our clinical observations could be additional features typical of the FOXP1S. Obviously, to corroborate this hypothesis and to better delineate the clinical spectrum of this rare syndromic form of NDD, the description of more patients carrying mutations in *FOXP1* and genotype–phenotype correlation studies are needed.

Regarding the genetic data, the variant c.1030C>T, p.(Gln344Ter) detected in the proband was never annotated before in literature or in a public database, and for this reason, it is useful for broadening the spectrum of deleterious *FOXP1* variants.

To date, a robust genotype-phenotype correlation has not been documented for FOXP1 syndrome. Different clinical aspects have been reported among affected subjects carrying the same recurrent variant. Furthermore, individuals in whom *FOXP1* gene deletion, truncating variants, and missense variants were found had no significant differences in the severity of global developmental delay [27]. Comprehensive assessment and longitudinal follow-up of individuals with *FOXP1* syndrome are necessary to better understand the clinical phenotype.

## 5. Conclusions

*FOXP1* haploinsufficiency is implicated in the etiology of an emerging syndromic form of NDD with a characteristic outline of marked speech and language delay. We describe a female patient carrying a novel heterozygous *FOXP1* mutation and clinical features never reported in medical literature that expand the clinical spectrum of *FOXP1*-syndrome. Obviously, to better elucidate the associated FOXP1S features and molecular mechanisms driving the onset of the syndrome, additional studies of individuals with ID of unknown cause are needed to identify novel *FOXP1* variants. From the perspective of affected families, a better understanding of the genetic basis of disease translates to more accurate prognosis, management, surveillance, and genetic advice, stimulates research into new therapies, and enables the provision of better support.

## Figures and Tables

**Figure 1 genes-14-01958-f001:**
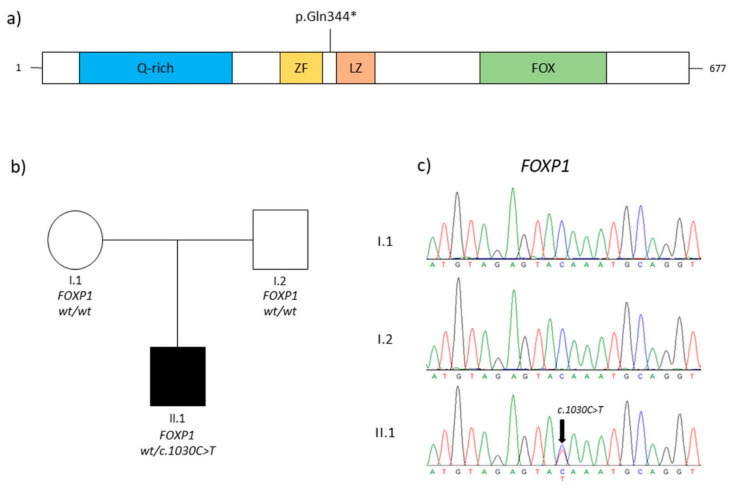
(**a**) Schematic representation of FOXP1 protein (FOXP1 domains are depicted as reported for Q9H334–1 in Uniprot; Q-rich, glycine-rich domain; ZF, zinc finger domain; LZ, leucine zipper domain; and FOX, DNA-binding forkhead box domain). The symbol * indicates a stop codon. (**b**) The pedigree of the family and the genotypes of individuals. Squares represent males, while circles represent females, respectively; clear and solid circles/squares indicate unaffected individuals and affected patients. (**c**) Electropherograms of the patient (II.1) and her parents (I.1, I.2). The black arrow indicates the mutation identified in this report.

**Table 1 genes-14-01958-t001:** Characteristics of the variant identified in the *FOXP1* gene.

Chromosome	Position	Reference Allele	Alternative Allele	Genotype	Gene	Nucleotide Change	Ammino Acid Change	dbSNP ID	TOPMED Allele Count	GnomAD ALL Allele Count
3	71050155	G	A	Het	*FOXP1* (NM_032682.6)	c.1030C>T	p.(Gln344Ter)	N.A.	N.A.	N.A.

Het, heterozygous; N.A., not available.

## Data Availability

The data presented in this study are reported in the Leiden Open Variation Databases (LOVD) (https://databases.lovd.nl/shared/variants/0000920404#00001788) (accessed on 1 March 2023).

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
