# Peer review of "Identification of a Novel FOXP1 Variant in a Patient with Hypotonia, Intellectual Disability, and Severe Speech Impairment"

_genes, 2023, doi:10.3390/genes14101958_

Round 1
Reviewer 1 Report
The manuscript provides a case report of a patient with a novel variant in FOXP1, exhibiting a phenotype that is broadly in keeping with the now well-described FOXP1 syndrome. Some new clinical features affecting the musculoskeletal and/or digestive system are described, which have not been reported previously to my knowledge, and which expand the phenotypic spectrum of the syndrome. Overall, it is a soundly conducted and well described report which makes a valuable contribution to the understanding of FOXP1 syndrome.
The manuscript would be improved by clarifying some aspects of the clinical description, including the methods used to assess certain clinical features. In particular, the report does not reference the recent detailed description of the FOXP1-related speech and language phenotype by Braden et al. (2021), which would provide valuable context for understanding the nature of the current patient’s speech and language impairment, and how it compares to other patients.
Comment 1
The description of the patient’s speech and language phenotype is brief and would benefit from more detail and some clarifications. A 2021 publication by Braden et al. (doi: 10.1111/dmcn.14955, PMID: 34109629) describes detailed cognitive, speech and language assessments of 29 patients with FOXP1 variants, and identifies a complex and distinct speech and language phenotype with dysarthria as the dominant feature. The authors should reference this publication and discuss how the current patient’s speech and language phenotype fits within the previously described phenotype. If any additional details about the patient’s speech and language phenotype (e.g. evidence of dysarthria, phonological deficits) are available, these should be added to the Clinical Description.
It would also be valuable to describe how the speech and language features were assessed for this patient. If these features were identified mainly by the clinician’s qualitative observations, this should be stated. Alternatively, if any standardised tests were used, these should be named and referenced.
Finally, the following statement should be clarified: "she spoke few words without meaning”. Does this mean that the patient used a small number of particular verbal utterances, but that these were not intelligible to the clinical examiners and/or to the parents? Or did the patient not consistently produce any speech at all? i.e. is the patient verbal but with low intelligibility, or is she largely non-verbal?
Comment 2
Related to Comment 1, but more broadly, the manuscript does not provide much information about how the clinical assessment was performed, or how certain clinical features were defined. These methods should be clarified, either in a new section in the Materials and Methods, or with added details in the Clinical Description. In particular, how were the following features defined? What criteria were used to define delay, or severe vs mild impairment?
• severe language delay
• intellectual disability
• motor delay, delayed psychomotor development
• severe speech impairment
Comment 3
I agree that the variant seems to be novel, though I was able to find a frameshift variant affecting the same amino acid residue (p.Gln344Hisfs*117) in ClinVar: https://www.ncbi.nlm.nih.gov/clinvar/variation/1457940/. The authors might consider briefly mentioning this other variant for additional context.
Comment 4
The authors have not conducted any functional evaluation of the variant at the transcript or protein level. This is justified, since it is a nonsense variant likely to result in nonsense mediated decay, with no truncated protein likely to be expressed in the patient’s cells. Thus, FOXP1 haploinsufficiency is the likely pathogenic mechanism. However I think this reasoning could be briefly stated in the manuscript for clarity.
Comment 5:
I was not able to find the data related to this manuscript in Array Express, as referenced in the Data Availability Statement. Depending on the journal’s policy, as well as the curation process at Array Express, this data should ideally be made publicly available before publication of the manuscript.
On the other hand, I’m not sure that Array Express is the best place to deposit this data, as it is generally focussed on high-throughput functional genomics. Since the authors have already reported the variant in the Leiden Open Variation Database, it would seem to make more sense to reference that submission in the Data Availability Statement.
Minor comments
1. Introduction, paragraph 1: This is a general overview of the FOXP protein family, yet the citation [reference 1] is for Song et al. - a case report of a single patient with a FOXP1 mutation. It would be more appropriate to cite a broader review of the FOXP proteins here.
2. Introduction, paragraph 4 to the end of section: Some instruction text from the journal seems to have been pasted here by mistake.
3. Materials and Methods, section 2.1: Suggest rewording from “tested on 1% electrophorese agarose gel" to “tested by electrophoresis on a 1% agarose gel”. I don’t think “electrophorese” is a common form of this word and may be confusing.
4. Results section 3.1. Clinical Description: The term "Caucasian" is rooted in pseudoscientific racial categories and should be avoided. A possible alternative could be "of European ancestry”. Ideally, the method of determining ancestry should also be included (i.e. self report vs genetic similarity analysis) For example: "She is the first child of healthy non-consanguineous parents of self-reported European ancestry.”
5. This sentence is ambiguous because 2 time points are mentioned: “At 16 months she sat independently, she spoke few words without meaning and began to take her first steps without support at 2.8 years.” Did she speak speak few words without meaning at 16 months, or at 2.8 years?
6. Consider changing “strabism” to “strabismus”, which is the more common form of this word in English.
7. “72th percentile” should read “72nd percentile”.
8. Results section 3.2. Genetic Analysis: there is a formatting error at the end of the paragraph: "3.1. Subsection" - probably part of a template that was not deleted.
9. Table 1: The column header "Amminoacid Change" should be "Amino Acid Change”.
10. Figure 1: In the legend for panel (c), “patiant” should be “patient”.
11. Discussion, paragraph 3: This sentence is unclear: "In particular, FOXP1 is expressed in the fetal telencephalon, interact with neural stem cells promoting their differentiation and migration through repression of the Notch signaling pathway [16]”.
12. Discussion, paragraph 3: “Reduction in the striatum and enlargement of the lateral ventricles has been observed in patients with FOXP1 haploinsufficiency [18-19]”. While enlargement of the lateral ventricles was reported in a human patient in reference [19] (Pariani et al.), the observation of reduction of the striatum in reference [18] (Bacon et al.) was in a mouse model. It should be made clear that the authors are citing a combination of human clinical observations and mouse model studies here.
13. Discussion, paragraph 3: "may result in abnormal development of motor pools [20]”. Should this say “motor skills”?
14. Discussion, paragraph 4: "unreveal" should be "reveal”.
15. Discussion, paragraph 7. There is some unclear phrasing: "The present case shares some of these features, including both some aspects regarding the neurobehavioral phenotype i.e. intellectual disabilities, hypotonia and severe speech impairment than facial dysmorphisms i.e. flat midface, widely spaced teeth, malformed ears”. Suggest changing “regarding” to “of”, and changing “than” to “and” or “as well as”.
16. Discussion, paragraph 10: References should be included for the following statements:
a) "Among cases with the same recurring variant different phenotypes have been described."
b) “In addition, individuals with FOXP1 deletions, truncating variants, and missense variants did not have significant difference in the severity of developmental delay."
17. Conclusions: Suggest changing “female” to “female patient”.
18. Conclusions: “at the best of our knowledge” should be “to the best of our knowledge”.
The English language in the manuscript is of high quality. I have pointed out a few minor typos and unclearly-worded sentences in the main Comments section.
Author Response
Author's Reply to the Review Report (Reviewer 1)
1# The description of the patient’s speech and language phenotype is brief and would benefit from more detail and some clarifications. A 2021 publication by Braden et al. (doi: 10.1111/dmcn.14955, PMID: 34109629) describes detailed cognitive, speech and language assessments of 29 patients with FOXP1 variants, and identifies a complex and distinct speech and language phenotype with dysarthria as the dominant feature. The authors should reference this publication and discuss how the current patient’s speech and language phenotype fits within the previously described phenotype. If any additional details about the patient’s speech and language phenotype (e.g. evidence of dysarthria, phonological deficits) are available, these should be added to the Clinical Description.
It would also be valuable to describe how the speech and language features were assessed for this patient. If these features were identified mainly by the clinician’s qualitative observations, this should be stated. Alternatively, if any standardised tests were used, these should be named and referenced.
Finally, the following statement should be clarified: "she spoke few words without meaning”. Does this mean that the patient used a small number of particular verbal utterances, but that these were not intelligible to the clinical examiners and/or to the parents? Or did the patient not consistently produce any speech at all? i.e. is the patient verbal but with low intelligibility, or is she largely non-verbal?
RESPONSE #1: As suggested, we referenced the publication by Braden et al. (doi: 10.1111/dmcn.14955, PMID: 34109629) and added in the Sections 3.1 and 4.0 details about the patient’s speech and language phenotype
Also, we added details on how the speech and language features were assessed for our patient in the Clinical Description.
Finally, to avoid confusion, we change the statement as follow: At the age of 16 months, the individual demonstrated the ability to sit independently. By 2,8 years old, she exhibited limited verbal communication with words lacking semantic meaning, and also started her first steps without external support.
2# Related to Comment 1, but more broadly, the manuscript does not provide much information about how the clinical assessment was performed, or how certain clinical features were defined. These methods should be clarified, either in a new section in the Materials and Methods, or with added details in the Clinical Description. In particular, how were the following features defined? What criteria were used to define delay, or severe vs mild impairment?
- severe language delay
- intellectual disability
- motor delay, delayed psychomotor development
- severe speech impairment
RESPONSE #2: Thanks a lot for your suggestion. We added details in the Clinical Description.
3# I agree that the variant seems to be novel, though I was able to find a frameshift variant affecting the same amino acid residue (p.Gln344Hisfs*117) in ClinVar: https://www.ncbi.nlm.nih.gov/clinvar/variation/1457940/. The authors might consider briefly mentioning this other variant for additional context.
RESPONSE #3: We appreciate your observation. We did not mention the p. (Gln344Hisfs*117) variant because, even if both involve the same amino acid, the molecular consequence are different (nonsense in our case, frameshift with a stop codon after 117 amino acids in the variant you consider). Additionally, Clinvar lacks information regarding the segregation study, so in that case is not possible to determine whether the condition is de novo, inherited from a mildly affected parent (mosaicism?), or inherited from a healthy parent who (a less likely scenario/germinal mosaicism).
4# The authors have not conducted any functional evaluation of the variant at the transcript or protein level. This is justified, since it is a nonsense variant likely to result in nonsense mediated decay, with no truncated protein likely to be expressed in the patient’s cells. Thus, FOXP1 haploinsufficiency is the likely pathogenic mechanism. However I think this reasoning could be briefly stated in the manuscript for clarity.
RESPONSE #4: We agree with the reviewer and added in the section 3.2 the sentence: “We did not perform any functional evaluation of the variant at protein level, since it is a nonsense variant likely to result in nonsense mediated decay, with no truncated protein likely to be expressed and detected in the patient’s cells”.
5# I was not able to find the data related to this manuscript in Array Express, as referenced in the Data Availability Statement. Depending on the journal’s policy, as well as the curation process at Array Express, this data should ideally be made publicly available before publication of the manuscript.
On the other hand, I’m not sure that Array Express is the best place to deposit this data, as it is generally focussed on high-throughput functional genomics. Since the authors have already reported the variant in the Leiden Open Variation Database, it would seem to make more sense to reference that submission in the Data Availability Statement.
RESPONSE #5: Thanks a lot for your suggestion. We changed it as suggested removing reference to array express
6# Introduction, paragraph 1: This is a general overview of the FOXP protein family, yet the citation [reference 1] is for Song et al. - a case report of a single patient with a FOXP1 mutation. It would be more appropriate to cite a broader review of the FOXP proteins here
RESPONSE #6: We have changed the citation to read: “Benayoun, B.A.; Caburet, S.; Veitia, R.A. Forkhead transcription factors: key players in health and disease. Trends Genet. 2011, 27, 224-232, doi: 10.1016/j.tig.2011.03.003.”
#7 Introduction, paragraph 4 to the end of section: Some instruction text from the journal seems to have been pasted here by mistake.
RESPONSE #7: We are sorry for the mistake. We removed this part.
8# Materials and Methods, section 2.1: Suggest rewording from “tested on 1% electrophorese agarose gel" to “tested by electrophoresis on a 1% agarose gel”. I don’t think “electrophorese” is a common form of this word and may be confusing.
RESPONSE #8: Thanks a lot for your suggestion. We changed it as suggested.
9# Results section 3.1. Clinical Description: The term "Caucasian" is rooted in pseudoscientific racial categories and should be avoided. A possible alternative could be "of European ancestry”. Ideally, the method of determining ancestry should also be included (i.e. self report vs genetic similarity analysis) For example: "She is the first child of healthy non-consanguineous parents of self-reported European ancestry.”
RESPONSE #9: Thanks you for your suggestion. We changed it as suggested
10# This sentence is ambiguous because 2 time points are mentioned: “At 16 months she sat independently, she spoke few words without meaning and began to take her first steps without support at 2.8 years.” Did she speak few words without meaning at 16 months, or at 2.8 years?
RESPONSE #10 Thank you for the suggestion. We changed the sentence to read: “At the age of 16 months, the individual demonstrated the ability to sit independently. By 2.8 years old, she exhibited limited verbal communication with words lacking semantic meaning, and also started her first steps without external support.”
11# Consider changing “strabism” to “strabismus”, which is the more common form of this word in English.
RESPONSE #11: We changed it as suggested
12# “72th percentile” should read “72nd percentile”.
RESPONSE #12: We changed it as suggested
13# Results section 3.2. Genetic Analysis: there is a formatting error at the end of the paragraph: "3.1. Subsection" - probably part of a template that was not deleted.
RESPONSE #13: We deleted it as suggested
14# Table 1: The column header "Amminoacid Change" should be "Amino Acid Change”.
RESPONSE #14: We changed it as suggested
15# Figure 1: In the legend for panel (c), “patiant” should be “patient”.
RESPONSE #15: We changed it as suggested
16# Discussion, paragraph 3: This sentence is unclear: "In particular, FOXP1 is expressed in the fetal telencephalon, interact with neural stem cells promoting their differentiation and migration through repression of the Notch signaling pathway [16]”.
RESPONSE #16: We removed the sentence to avoid confusion.
17# Discussion, paragraph 3: “Reduction in the striatum and enlargement of the lateral ventricles has been observed in patients with FOXP1 haploinsufficiency [18-19]”. While enlargement of the lateral ventricles was reported in a human patient in reference [19] (Pariani et al.), the observation of reduction of the striatum in reference [18] (Bacon et al.) was in a mouse model. It should be made clear that the authors are citing a combination of human clinical observations and mouse model studies here.
RESPONSE #17 We agree with the observation. We changed the sentence as follow: “FOXP1 is also expressed in the developing striatal projection neurons and basal ganglia [5-7], and reduction in the striatum has been observed in FOXP1 haploinsufficient mouse models [8], while enlargement of the lateral ventricles has been observed in human patients with FOXP1 haploinsufficiency [9]
18# Discussion, paragraph 3: "may result in abnormal development of motor pools [20]”. Should this say “motor skills”?
RESPONSE #18: Yes, thank you for your observation. We changed it as suggested.
19# Discussion, paragraph 4: "unreveal" should be "reveal”.
RESPONSE #19: Yes, thank you for your observation. We changed it as suggested.
20# Discussion, paragraph 7. There is some unclear phrasing: "The present case shares some of these features, including both some aspects regarding the neurobehavioral phenotype i.e. intellectual disabilities, hypotonia and severe speech impairment than facial dysmorphisms i.e. flat midface, widely spaced teeth, malformed ears”. Suggest changing “regarding” to “of”, and changing “than” to “and” or “as well as”.
RESPONSE #20: Yes, thank you for your observation. We changed it as suggested
21# Discussion, paragraph 10: References should be included for the following statements:
- a) "Among cases with the same recurring variant different phenotypes have been described."
- b) “In addition, individuals with FOXP1 deletions, truncating variants, and missense variants did not have significant difference in the severity of developmental delay."
RESPONSE #21: We appreciate your suggestion. We added citation “24 - Meerschaut, I. Rochefort, D. Revençu, N. Pètre, J. Corsello, C. Rouleau, G.A. Hamdan, F.F. Michaud, J.L. Morton, J. Radley, J. et al. FOXP1-related intellectual disability syndrome: a recognisable entity. J Med Genet. 2017, 54, 613-623. doi: 10.1136/jmedgenet-2017-104579”.
Both a) and b) observation are cited by [24]
22# Conclusions: Suggest changing “female” to “female patient”.
RESPONSE #22: Yes, thank you for your observation. We changed it as suggested
23# Conclusions: “at the best of our knowledge” should be “to the best of our knowledge”.
RESPONSE #23: Yes, thank you for your observation. We changed it as suggested

Reviewer 2 Report
This study could be useful to add knowledge about a new syndrome. Unfortunately, the novelty of this case report compared to the patients reported in the literature is limited.
-it could be helpful to well describe the clinical phenotype of this child. The genu valgum is associated with hyperlaxity? scoliosis?
-absence of sphincter control up to 5 years: your last examination is at 10 years old. She has reached the sphincter control? Also, I would suggest that the patient reported from Cesaroni et al (2023) had also only a partial control of sphincter at 4 years old
-Let's put the name of the gene in italics
- line194-202: you mentioned only the study of Lozano et al (2021). You could add the most recent case reports and compare your patient with also these patients
-line 52-66: eliminate this part
Author Response
Author's Reply to the Review Report (Reviewer 2)
1# -it could be helpful to well describe the clinical phenotype of this child. The genu valgum is associated with hyperlaxity? scoliosis?
RESPONSE #1: No, in our patient, it is not associated with hyperlaxity or scoliosis
2# absence of sphincter control up to 5 years: your last examination is at 10 years old. She has reached the sphincter control? Also, I would suggest that the patient reported from Cesaroni et al (2023) had also only a partial control of sphincter at 4 years old
RESPONSE #2: At last clinical evaluation the patient showed only a partial control of sphincter
3# Let's put the name of the gene in italics
RESPONSE #3: Sorry for the mistake, we have corrected the format of the gene name.
4# line194-202: you mentioned only the study of Lozano et al (2021). You could add the most recent case reports and compare your patient with also these patients
RESPONSE #4: Thank you for your suggestion. We added in the paper the case report by Cesaroni et al (2023), which recently report a further case, provide previously unappreciated clinical features associated with FOXP1-related intellectual disability (ID) syndrome and summarise the clinical feature of the syndrome.
5# line 52-66: eliminate this part
RESPONSE #5: We are sorry for the mistake, we removed this part.

Reviewer 3 Report
Dear authors, congratulations on your paper! I find it very interesting. FOXP1 syndrome is indeed a rare disease, and every case-report is worth publishing. However, I have a few recommendations, which I believe would improve the manuscript.
Introduction section - could be expanded. Lines 166-193 from your discussion sound more like introduction and could be moved to Introduction section. I believe you did not delete some sentences form the template file - lines 52-66, page 2.
Materials and methods section - the work protocol is very well described. Lines 106-107, page 3 - The resulting putative pathogenic variants were confirmed by Sanger sequencing in both the proband and the parents’ DNA. - The parents were not carriers and this was a de novo mutation, please, correct this sentence.
Results section - the results were presented in a clear manner. The figures were great.
Discussion section - this section includes a lot of basic information, which could be part of the Introduction. You should include more information about other published cases. For example, Chang et al. published a paper about a patient with a heart defect similar to yours. There is an interesting study by Wang et al. on mitochondrial dysfunction, oxidative stress and FOXP1 mutations in animal models. These are just some examples, I am sure there are other papers as well.
Conclusion section - a good conclusion, no recommendations.
Overall, the English language needs to be improved, for example - ... best know member, should be known; severe language disorder were..., should be was; these reason 215 useful to expand the molecular spectrum of pathogenic FOXP1 mutations, should be that reason, etc.
Author Response
Author's Reply to the Review Report (Reviewer 3)
1# Introduction section - could be expanded. Lines 166-193 from your discussion sound more like introduction and could be moved to Introduction section. I believe you did not delete some sentences form the template file - lines 52-66, page 2
RESPONSE #1: Thanks a lot for your suggestion. We modify the paper as suggested.
2# Materials and methods section - the work protocol is very well described. Lines 106-107, page 3 - The resulting putative pathogenic variants were confirmed by Sanger sequencing in both the proband and the parents’ DNA. - The parents were not carriers and this was a de novo mutation, please, correct this sentence.
RESPONSE #2: We correct the sentence as suggested. To avoid confusion, the new one is: “Confirmation and segregation analysis of the putative pathogenic variant were performed by Sanger sequencing on the proband’s and parents’ DNA”. Also, in the section 3.2 we state: “Parental DNA analysis showed that it is a de novo event (Figure 1).”
3# Discussion section - this section includes a lot of basic information, which could be part of the Introduction. You should include more information about other published cases. For example, Chang et al. published a paper about a patient with a heart defect similar to yours. There is an interesting study by Wang et al. on mitochondrial dysfunction, oxidative stress and FOXP1 mutations in animal models. These are just some examples, I am sure there are other papers as well
RESPONSE #3: Thanks a lot for your suggestion. We have moved some part of discussion on introduction and considered the paper by Chang et al.
Regarding the paper suggested, we know the excellent work of Wang and colleagues in which through studies on animal models they demonstrate how a disrupted mitochondrial network and the resulting oxidative stress in the hippocampus contribute to the altered learning and cognitive impairment in Foxp1+/- mice suggesting that the same mechanism may underlie some clinical manifestations observed in patients affected by FOXP1S but we did not mention it simply because the purpose of our study is to report a new variant of FOXP1, useful for expanding the spectrum of known pathogenetic mutations and clinically describing a new patient to further delineate the phenotype associated with the syndrome.
